# Mechanisms of Primary and Acquired Resistance to Immune Checkpoint Inhibitors in Patients with Hepatocellular Carcinoma

**DOI:** 10.3390/cancers14194616

**Published:** 2022-09-23

**Authors:** Stefania De Lorenzo, Francesco Tovoli, Franco Trevisani

**Affiliations:** 1Oncology Unit, Azienda USL Bologna, 40124 Bologna, Italy; 2Department of Medical and Surgical Sciences, University of Bologna, 40126 Bologna, Italy; 3Division of Internal Medicine, Hepatobiliary and Immunoallergic Diseases, IRCCS Azienda Ospedaliero-Universitaria di Bologna, 40138 Bologna, Italy; 4Unit of Semeiotics, Liver and Alcohol-Related Diseases, IRCCS Azienda Ospedaliero-Universitaria di Bologna, 40138 Bologna, Italy

**Keywords:** liver cancer, hepatocellular carcinoma, immunotherapy, immune checkpoint inhibitors, atezolizumab, bevacizumab, tremelimumab, durvalumab, tumour microenvironment, resistance, cirrhosis, outcome

## Abstract

**Simple Summary:**

Immune checkpoint inhibitors (ICIs) are now a cornerstone of systemic treatment for hepatocellular carcinoma (HCC). However, their efficacy is blunted by mechanisms of tumour resistance in many patients. This review reports on the state of the art of resistance to ICIs, focusing on HCC, with the aim to provide clear and direct information to clinicians and researchers. Growing knowledge on the mechanisms of resistance to immunotherapy can indeed guide the choice of and expand the application of novel combined treatments.

**Abstract:**

Hepatocellular carcinoma (HCC) is the most common liver cancer and a relevant global health problem. Immune checkpoint inhibitors (ICIs) represent the most effective systemic treatment for HCC. However, due to primary resistance, approximately 40% of HCC patients do not achieve a disease control with ICIs. Moreover, a similar proportion will experience disease progression after an initial response caused by secondary resistance. This review describes the mechanisms of primary and secondary resistance and reports the ongoing therapeutic strategies to overcome these obstacles.

## 1. Introduction

In oncology, the main goal of immunotherapy is to induce and boost the ability of immune cells to destroy cancer cells, viewed as an “altered self”. This goal can be achieved using different strategies, including the use of immune checkpoint inhibitors (ICIs). They are key modulators of the T-cell-mediated antitumour immune response, and their interaction can activate either inhibitory or activating immune signalling pathways [1,2]. The advent of immunotherapy has revolutionized the therapeutic approach to many tumours, including hepatocellular carcinoma (HCC), the most common liver cancer and a relevant global health problem [3]. Two ICI-based different therapeutic combinations (atezolizumab–bevacizumab and tremelimumab–durvalumab) have been proven to improve the overall survival (OS) of HCC patients compared to sorafenib [4,5], a multitarget tyrosine kinase inhibitor (TKI) which has represented the mainstay of treatment for HCC since 2008 [6].

Currently approved ICIs for HCC include monoclonal antibodies directed against cytotoxic T lymphocyte-associated protein 4 (CTLA-4), programmed cell death protein 1 (PD-1) and programmed cell death ligand 1 (PD-L1). A further type of ICI, acting through the inhibition of lymphocyte-activation gene 3 (LAG-3), has shown promising results in melanoma [7] and is under investigation for HCC.

Despite the success of ICIs in the treatment of several tumours, most patients develop a resistance to ICIs. This resistance may be either primary (lack of initial response) or secondary (progression following initial disease control). The resistance can be further classified as intrinsic, if elicited by the tumour itself, or extrinsic, if due to the interaction with normal stromal cells in the tumour microenvironment (TME) [8,9,10].

This review analyses the general mechanisms of primary and secondary resistance to ICIs occurring in immune oncology, also addressing the HCC-specific mechanisms of this phenomenon. Moreover, ongoing therapeutic strategies to tackle and overcome resistance, as well as open questions and future perspectives are presented.

## 2. Tumour Immunity Cycle

Our immune system is a complex system that operates through two main modalities of response: innate immunity (a non-specific and quick response) and adaptive immunity (subsequently activated antigen-specific response that mediates immune memory) [11].

Although a number of cellular components are involved in this complex system, antitumour immune surveillance is based on T lymphocytes, as demonstrated by several mice models, in which the loss of T-cell activity was associated with uncontrolled tumour growth. T lymphocytes recognise and identify some molecules released from cancer cells as foreign (or “antigens”). These tumour-associated molecules are mutated proteins, showing alterations caused by the high genetic instability of tumours. Indeed, the high degree of genetic diversity in cancer cells accelerates their evolutionary fitness but also increases the divergence from a normal cell, facilitating recognition by the immune system. The frequency of non-synonymous somatic mutations per DNA megabase in the coding genome of a tumour is called “tumour mutational burden” (TMB) [12,13].

Overall, activating antitumour immunity is a multistep process involving several lymphocyte populations. The uncontrolled growth and the consequent high TMB of cancer cells firstly result in the activation of innate immunity, including natural killer (NK) cells, which target cancer cells and favour their apoptosis leading to the release of tumour-associated antigens. These molecules are released into the tumour microenvironment (TME) and are subsequently recognised by antigen-presenting cells (APCs), which capture these neo-antigens and present them to the corresponding T-cell receptor (TCR) of naïve T lymphocytes via the major histocompatibility complex (MHC).

The final step of these immune events is the release of co-stimulatory cytokines by CD4+ T cells and the activation of antigen-specific CD8+ T cells leading to the lymphocyte-mediated destruction of tumour cells (Figure 1) [14,15].

## 3. Tumour Immune Escape

In acute infections, T cells can eliminate the pathogen. Instead, in the case of advanced tumours, the neo-antigen is rarely eliminated and, consequently, a chronic inflammatory stimulation is established with the possibility of losing effectiveness. In fact, chronic immune stimulation activates negative feedback mechanisms (in particular, the activation of immunosuppressive cells that silence the immune response) and promotes a progressive loss of the cytotoxic capacities of T cells.

On the other hand, cancer cells can develop a series of resistance mechanisms aimed at evading the host’s immune control. Namely, tumour cells can acquire the ability to express molecules that inhibit the immune response or reduce the expression of the MHC complex in APCs [16,17]. The most known immune checkpoint molecules are CTLA4, PD-1 and its ligand PD-L1, and lymphocyte-activation gene 3 (LAG-3). All of them are receptors expressed on the membrane of T cells in different phases. They counterbalance lymphocyte activation in physiological conditions to avoid unnecessary tissue damage, chronic inflammation, and uncontrolled lymphocytic proliferation.

In detail, CTLA-4 is expressed by T cells in the priming phase, exerting inhibitory actions following interaction with its ligands, CD80 (B7-1) and CD86 (B7-2), which are expressed on the surface of the APCs. PD-1, instead, is expressed in the effector phase, especially under conditions of chronic exposure to the antigen, and exerts an inhibitory action on the lymphocytes via interaction with its ligands, PD-L1 and PD-L2, found in the TME. Similarly, LAG3 is expressed in lymphocytes during the effector phase [18]. LAG3’s most important ligand is represented by the MHC class II molecules expressed on the surface of APCs, with an interaction ultimately leading to T-cell exhaustion [19].

Tumours can exploit these mechanisms to avoid immune-mediated destruction not only by expressing ligands activating the immune checkpoints themselves but also favouring the development of a tolerant TME through the recruitment of non-neoplastic cells expressing these ligands.

Immune checkpoint inhibitors (ICI) are monoclonal antibodies specifically designed to disrupt these ligand/receptor interactions, removing the inhibition of T cells and promoting their antitumoral cytotoxic activity (Table 1) [20,21,22].

## 4. Mechanisms of Primary ICI Resistance

### 4.1. Tumour-Intrinsic Mechanisms

Tumour intrinsic mechanisms include a lack of tumour immunogenicity (low TMB, heterogeneous antigens, mutation of critical genes involved in immune regulation), defective antigen presentation, and aberrations in several signalling pathways (Figure 2).

Low TMB (currently defined as <10 muts/Mb, using a pan-cancer threshold) [12,13] is one of the principal mechanisms of primary resistance to immune therapies. Tumours with the highest number of somatic mutations, being enriched in neo-antigens, are therefore potentially more immunogenic. Accordingly, the best response rate to ICIs can be observed in tumours with high TMB, such as melanoma and non-small cell lung cancer (NSCLC). In patients bearing these cancers with high TMB, ICI treatment was associated with an improved overall survival (OS), while patients harbouring neoplasms with low TMB showed a lower objective response rate to ICIs and a poorer OS [23,24,25]. However, most patients did not respond to immunotherapy regardless of their TMB status, suggesting that the presence of high tumour somatic mutations is only one factor contributing to the immune response.

Another mechanism of ICI resistance is the dysfunction of neo-antigen presentation. As previously stated, tumour cells produce neo-antigens that can be recognised by the MHC system. To avoid identification by T cells, tumour cells can decrease the expression of neo-antigens, such as by the hypermethylation of genes expressing these antigens [13]. Moreover, the activation of anti-cancer-specific CD8+ T cells requires an efficient presentation of tumour antigens. Cancer cells can develop mutations in genes associated with MHC presentation, such as beta-2 microglobulin, leading to the severe dysfunction of the antigen-presenting mechanism and resistance to ICIs [26,27,28].

Loss-of-function mutations in the Janus tyrosine kinases 1 and 2 (JAK-1 and JAK-2) is another cause of intrinsic resistance to ICIs. The integrity of these kinases is essential for interferon-alfa (IFN-α) intracellular signalling. Since the INF-α pathway plays a vital role in the priming of T cells by APCs, a defective JAK1/2 pathway efficiently inhibits the activation of effector T cells [8,10].

Additionally, the WNT/β-catenin pathway promotes the resistance to ICIs by the transcriptional repression of key chemokine genes, leading first to a depletion of basic leucine zipper ATF-like transcription factor 3 (Batf3)-lineage DC recruitment, and then to a consequent failure to prime and recruit CD8+ T cells [29].

The WNT/β-catenin pathway is of particular interest in the case of HCC, as approximately 40% of these tumours show the constitutive activation of WNT/β-catenin signalling induced by relevant gene mutations [30,31]. Most interestingly, constitutive WNT/β-catenin activation had negative effects both on the disease control rate and progression-free survival (PFS) of HCC patients who received anti-PD1 therapies [32,33].

The phosphatase and tensin homologue-signal transducer and activator of transcription 3 (PTEN-STAT3) pathway is another crucial pathway in immune regulation. PTEN knock-down decreases the ability of T cells to kill tumour cells through the indirect activation of STAT3, a transcription factor that is a key signalling node and a regulator of the critical hallmarks of cancer, tumour angiogenesis, resistance to apoptosis, metastasis, and immune evasion [34]. In particular, STAT3 activation has been linked to inhibitory effects on immune response, including the down-regulation of triggering receptor expressed on myeloid cells-1 (TREM-1) and the reduced expression of Toll-like receptors and interferon-inducible genes [35]. Nearly 60% of human HCCs exhibit activated nuclear STAT3, and STAT3 activation is associated with a poor prognosis [35].

### 4.2. Tumour-Extrinsic Mechanisms

Tumour-extrinsic mechanisms are mainly due to the immune-suppressive effect of TME. They include primary T-cell-related factors (activation of alternative immune checkpoints, T-cell exhaustion and phenotype alteration), immunosuppressive cells, cytokines and metabolites released into the TME (Figure 3). As the TME includes immune cells, blood vessels, fibroblasts, signalling molecules, and the extracellular matrix surrounding tumour cells, many factors can contribute to the development of an immune-suppressive TME [36].

Infiltration of the TME by unique types of immune cells, including myeloid-derived suppressor cells (MDSCs) and regulatory T cells (Tregs), can suppress T-cell recruitment and responses. MDSCs and Tregs are both immune cells, but derive from different lineages (myeloid and lymphoid, respectively). Under physiological conditions, both populations are involved in the modulation of immune response, thereby maintaining homeostasis and self-tolerance [36,37]. This aim is achieved through the release of inhibitory cytokines (i.e., transforming growth factor beta—TGF-β—or IL-10) or the up-regulation of immune checkpoints (such as CTLA4 and PD-1) [36,37]. Alternative co-inhibitory immune checkpoints on T cells, including T-cell immunoglobulin, can also be up-regulated, inducing T-cell exhaustion by activating the apoptosis pathways [9]. High tumour infiltration by MDSCs has been correlated with poor prognosis and resistance to ICI therapy, as they can release metabolic enzymes inhibiting T-cell expansion and promoting the conversion of naïve T cells into Tregs [36].

LAG-3 is an example of the activation of immune checkpoints outside of the classical PD-1/PD-L1 axis. Its expression has been associated with a poor prognosis in patients with HCC [38]. Furthermore, T-cell immunoglobulin and mucin-containing protein-3 (TIM-3) and its ligand galectin-9 can activate a complex cascade, ultimately leading to T-cell exhaustion [38]. LAG-3, TIM-3 and PD-1 can act synergistically, facilitating HCC immune evasion and mediating the resistance to classical PD-1/PD-L1 blockades [39,40].

Under different circumstances, TME-mediated resistance to ICIs might not depend on immune-derived cells primarily. Epithelial-to-mesenchymal transition (EMT) is a cellular process enabling epithelial cells to gain mesenchymal features leading to an aggressive and motile phenotype [41]. EMT can promote an immunosuppressive TME via the recruitment of APCs, inducing tolerance, up-regulation of the immune checkpoint, and resistance to NK cell-mediated lysis [42,43]. The association between EMT, immunosuppression, tumour progression and metastasis has been reported in different cancer types, including HCC [44].

On most occasions, a very complex interplay between immune and non-immune cell resistance is the cause of resistance. The VEGF and TGF-β pathways are master regulators of these processes. The VEGF signalling pathway is a well-known regulator of neo-angiogenesis production by HCCs. It can also produce immune resistance via the induction of the Fas ligand, resulting in the death of tumour-infiltrating CD8+ T cells [45]. TGF-β is a multifunctional cytokine with a multitude of effects on immune and stem cell differentiation and regulation [46,47]. In many cancer cells, the disruption of TGF-β signalling causes the simultaneous proliferation of both cancer cells and the surrounding stromal cells in an immunosuppressive and pro-angiogenic microenvironment [48]. Additionally, TGF-β can convert effector T cells into regulatory T cells [49] and exert inhibitory effects on B cells [50], turning off the inflammatory reaction and favouring tumour immune escape.

Last but not least, even the composition of extracellular matrix molecules can influence the recruitment of immune-suppressor cells or exclude T-cell infiltration. In breast cancer studies, T cells easily migrated in a loose collagen matrix, while a dense collagen matrix hampered T-cell migration [51,52]. Indoleamine-pyrrole 2,3-dioxygenase 1 (IDO-1) is a heme-containing enzyme in the extracellular matrix that can interfere with immune clearance. IDO-1 is physiologically expressed in many tissues and cells, but its activation can also help malignant cells escape immune clearance [36]. Tumours with high IDO1 can deplete the essential amino acid tryptophan from the TME, resulting in T-cell anergy and immune suppression [53].

## 5. Mechanisms of Acquired ICI Resistance

Although the specific mechanisms of acquired (secondary) resistance are poorly understood, some of these are overlapped in primary and acquired ICI resistance. In the specific case of HCC, these mechanisms are still elusive. For this reason, we will report information derived from studies in other malignancies.

Tumour cells can lose some encoding sequences for tumour neo-antigens through the elimination of specific sub-clones or the deletion of chromosomal regions, resulting in the selection of less immunogenic tumour cell clones spared from T-cell killing [54,55]. For example, an acquired homozygous loss of beta-2 microglobulin, decreasing the expression of HLA and disrupting the antigen presentation, was identified in lung cancer. Defective HLA class I antigen processing through deleterious mutations in beta-2 microglobulin has been shown in melanoma [56,57]. Regarding melanoma, acquired resistance to PD-1 inhibitors can also be mediated by JAK1/2-inactivating mutations.

The up-regulation of both immune checkpoints and immune-suppressive cytokines may be other causes of acquired resistance. For instance, lung cancer patients with acquired resistance against anti-PD1 therapy show an up-regulation of PD-1 [9]. Although it is still unclear whether PD-L1 expression causes an increased or decreased ICI response, it could result in the suppressed antitumour effect of T cells [58].

## 6. Current Strategies in Immunotherapy for HCC

The initial studies of immunotherapy in HCC patients included ICI monotherapies [59]. Despite promising results of Phase 2 trials, two different Phase 3 randomised clinical trials failed to show the superiority of nivolumab vs. sorafenib in the frontline setting, and pembrolizumab vs. placebo in the second line setting, respectively [60,61]. Nevertheless, trials of ICI monotherapies provided three pieces of important information: First, these regimens were safe and well-tolerated, even in patients with liver cirrhosis and chronic viral infections. Second, ICIs obtained an objective response rate in 15–18% of cases [60,61], which is considerably higher that of TKI. Third, patients with an objective response had very long survival times [62].

Therefore, associating ICIs with agents able to overcome resistance to immunotherapy has become the most common strategy to improve the response rate and survival of HCC patients (Figure 4).

The following paragraphs describe these combinations, reporting both established strategies (i.e., combinations of drugs tested in concluded Phase 3 trials) and novel perspectives.

### 6.1. Established Strategies

#### 6.1.1. Anti PD1/PD-L1 in Association with Intravenous Anti-VEGF Agents

The IMBrave150 study was a global Phase 3 clinical trial in which 501 patients were randomly assigned to the combination of atezolizumab–bevacizumab or sorafenib according to a 2:1 ratio [4]. The scientific rationale for this combination relies on the knowledge that the VEGF blockade can transform the immunosuppressive TME into an immunostimulatory milieu. In particular, bevacizumab may normalise the aberrant vascular–immune crosstalk by reorganising malformed tumour vessels to improve the infiltration of CD8+ T and CD4+ TH1 cells into the TME [63].

When the IMBrave150 trial was stopped at an interim analysis due to the manifest superiority of the combination treatment arm, atezolizumab–bevacizumab became the new frontline strategy for the systemic treatment of HCC, ending the 14-year-long era of sorafenib. In fact, the median OS was not reached in the atezolizumab–bevacizumab arm after 17 months, while it occurred at 13.2 months in the sorafenib arm [4]. Moreover, after a longer follow-up, the median OS in the atezolizumab–bevacizumab group was 19.2 months, the longest ever achieved with systemic therapy for HCC patients (95% CI 17.0–23.7) [64]. The median PFS was 6.8 months (95% confidence interval 5.7–8.3) vs. 4.3 months (95% confidence interval, 4.0–5.6) in the combination and sorafenib arms, respectively. The ORR was also significantly higher in the atezolizumab–bevacizuamb arm compared with the sorafenib arm (27.3% vs. 11.9%, *p* < 0.001). Overall, the toxicity of the combined therapy was manageable. The most common AE was arterial hypertension (15%), a known bevacizumab-related AE [4]. Serious AEs were slightly higher in the combination than in the sorafenib arm (38.0% vs. 30.8%), and the same goes the proportion of patients that permanently discontinued the treatment due to toxicity (15.5% vs. 10.3%). However, the combined therapy delayed the time point of clinical deterioration and was associated with a better quality of life than sorafenib [4].

Interestingly, real-life data on atezolizumab–bevacizumab would confirm if the concurrent VEGF blockade can overcome some mechanisms jeopardising the response to anti-PD1/PD-L1 monotherapy, as patients with WNT/β catenin mutations had brilliant responses to the combination treatment [65,66].

#### 6.1.2. Combination of PD-1 and CTLA-4 Inhibitors (Dual-Checkpoint Blockade)

Adding the CTLA-4 to the PD-1/PD-L1 blockade can enhance the immune response against the tumour, blocking a second immune checkpoint exploited by tumour cells to evade immune surveillance.

This strategy has been adopted the HIMALAYA study, a Phase 3 trial comparing the association of tremelimumab and durvalumab vs. sorafenib in the frontline setting. In this trial, the ICI combination was administered as a single, high-priming dose of tremelimumab plus durvalumab, followed by a cadenced infusion regimen of durvalumab, termed STRIDE (single-tremelimumab regular-interval durvalumab). The primary outcome was OS for STRIDE vs. sorafenib. Superiority and noninferiority for durvalumab alone vs. sorafenib were secondary outcomes [5].

A total of 1171 patients were enrolled and randomly assigned to STRIDE, durvalumab or sorafenib. The median OS was 16.4 months (95% CI 14.2–19.6) with STRIDE, 16.6 months (95% CI 14.1–19.1) with durvalumab, and 13.8 months (95% CI 12.3–16.1) with sorafenib [5]. The OS with STRIDE was superior to sorafenib (hazard ratio 0.78, 96% CI 0.65–0.90), and durvalumab monotherapy was noninferior to sorafenib (hazard ratio 0.86; 96% CI 0.73–1.03; noninferiority margin, 1.08). Median PFS was not significantly different among the three groups. Objective response rates per investigator assessment were 20.1% with STRIDE, 17.0% with durvalumab, and 5.1% with sorafenib [5].

A second Phase 3 trial (CheckMate 9DW, NCT04039607) is ongoing, comparing nivolumab–ipilimumab vs. sorafenib in HCC patients. This trial is the natural continuation of a previous Phase 2 study which documented an impressively high overall response rate (31%) with a manageable safety profile [67].

#### 6.1.3. Anti PD1/PD-L1 in Association with TKIs

The combination of ICIs with TKIs has been considered another strategy to treat HCC with immunotherapy to ameliorate the results of each monotherapy. Similarly to intravenous anti-VEGF agents, TKIs could have synergistic effects as they can block the signalling from various growth factors and affect immune effectors of the tumour vasculature [68]. Unfortunately, two Phase 3 RCTs failed to demonstrate the superiority of such combinations compared to TKI monotherapy.

In the COSMIC-312 trial, 837 patients were randomised in three arms to receive sorafenib, cabozantinib, or a combination of atezolizumab + cabozantinib. The dual primary endpoints were PFS and OS. The median PFS was 6.8 months in the combined treatment group vs. 4.2 months in the sorafenib group (HR 0.63, 99% CI 0.44–0.91]. The median OS was 15.4 months (96% CI 13.7–17.7) in the combined treatment group, and 15.5 months (12.1–not estimable) in the sorafenib group (HR 0·90, 96% CI 0.69–1.18) Partial response was achieved in 12% and 4% of patients in the atezolizumab–cabozantinib and sorafenib groups, respectively [69]. Even if PFS and OS were co-primary endpoints (and, therefore, the study technically succeeded), the lack of an OS advantage in a setting in which the atezolizumab–bevacizumab combination had already achieved a clear advantage over the same competitor makes any further commercial progression of atezolizumab–cabozantinib unlikely.

A very recent press release reported the failure of another combination between ICIs and TKIs in the LEAP-002 study [70]. In this Phase 3 RCT, all patients were treated with lenvatinib and were randomised 1:1 to receive either pembrolizumab or placebo as an additional treatment. Even if trends toward improvement in OS and PFS were observed in patients who received the combined treatment, these results did not meet statistical significance, as the median OS of the lenvatinib monotherapy arm was longer than that observed in previously reports regarding unresectable HCC [70].

Two other Phase 3 trials testing combinations of anti-PD1/PD-L1 and TKIs vs. a TKI monotherapy (NCT04194775, NCT04344158) are currently ongoing. Their results will help understand whether this strategy is still viable, especially in light of successful trials exploring other combinations.

### 6.2. Novel Strategies

A number of clinical trials are investigating ICIs in combination with drugs that would prevent resistance to immunotherapy and counteract the tumour-intrinsic and -extrinsic mechanisms of resistance.

#### 6.2.1. Combination with Drugs Acting on Tumour-Intrinsic Mechanisms of Resistance

In the case of intrinsic resistance, relatlimab is the leading therapeutic agent to discuss. It is a first-in-class LAG-3 inhibitor which can restore the effector function of exhausted T cells [71]. The synergistic effect of the dual PD-1/LAG-3 blockade has been tested in preclinical models and in a recent global Phase 2–3 trial in patients with melanoma. The RELATIVITY-047 trial enrolled 714 patients with untreated melanoma who were randomised 1:1 to receive either the standard-of-care nivolumab monotherapy or a combination of nivolumab + relatlimab, with PFS as the primary endpoint [72]. The combined treatment arm demonstrated a superiority, with a median PFS of 10.1 vs. 4.6 months (hazard ratio 0.75, 95% confidence interval 0.62–0.92) [72]. The nivolumab–relatlimab combination is currently under investigation as a second-line strategy for HCC patients who received TKI as first-line agent in the Phase 2 RELATIVITY-073 trial (NCT04567615).

Agonists of the NK cells are other attractive agents, as they can restore the NK-mediated clearance of tumour cells trying to evade the immune response by modifying the MHC molecules [59]. A Phase 1–2 trial (NCT03897543) is testing ABX-196, an NK agonist, in combination with nivolumab in HCC patients who have received at least one prior TKI treatment.

DKN-01 is a monoclonal antibody neutralising DKK1, an immune-suppressive protein produced by tumours harbouring Wnt/β catenin activation. A recent Phase 2a trial exploring the effect of DKN-01 in combination with chemoimmunotherapy showed encouraging results in patients with advanced gastroesophageal adenocarcinoma [73]. DKB-01 is also under investigation in a Phase 1–2 trial (NCT03645980) as a potential treatment of HCC with Wnt-β catenin activation. This study was designed in 2018 when sorafenib was still the standard of care, so DKB-01 is administered in association with sorafenib. Future studies evaluating DKB-01 in combination with ICIs in HCC patients cannot be excluded.

The immune-suppressive effects derived from the activation of the PTEN-STAT3 pathway are also being targeted. TTI-101 is a first-in-class, orally bioavailable, selective small molecule that prevents STAT3-mediated transcriptional activity [34]. TTI-101 has demonstrated antitumour activity across a broad range of preclinical cancer models, including a HepPten (hepatocyte-specific deletion of Pten) murine model of liver cancer which recapitulates the pathogenesis of HCC in non-alcoholic fatty liver disease (NAFLD) with chronic inflammation and liver fibrosis leading to cancer [34]. For this reason, the Food and Drug Administration has granted orphan drug designation to TTI-101 for HCC [74]. Additionally, a Phase 1–2 study of TTI-101 in monotherapy or in combination with pembrolizumab or atezolizumab–bevacizumab is currently ongoing (NCT05440708).

Finally, resistance to the ICI-mediated attack on tumour cells could be prevented by inhibiting autophagy, a mechanism by which cancer cells evade apoptosis [75]. Ezurpimtrostat is a quinoline derivative exerting an in vitro antiproliferative activity that blocks autophagic activity and the relocalisation of mTOR in HCC cells [76]. Preliminary in vivo results of an ezurpimtrostat monotherapy showed an excellent tolerance but absent objective responses [77]. Consequently, ezurpimtrostat is now being evaluated in association with atezolizumab–bevacizumab in the ABE-LIVER Phase 2 trial (NCT05448677).

#### 6.2.2. Combination with Drugs Acting on Tumour-Extrinsic Mechanisms of Resistance

The TGF-beta pathway has been a target for HCC drugs for a long time, since its inhibition leads to both a reduction in the EMT and a reactivation of NK cells. The oral TGF-beta inhibitor galunisertib had been tested as a possible treatment for HCC in combination with nivolumab since 2015, but the manufacturer discontinued its development in January 2020 [78]. Instead, the monoclonal anti-TGF-β antibody ascrinvacumab showed promising results for HCC patients in a Phase 1–2 trial [79] and is currently under investigation as a second-line treatment in a Phase 2 trial in combination with nivolumab (NCT05178043).

The modulation of suppressive immune cells is becoming a novel target. SD-101 is a Toll-like receptor 9 agonist binding to the receptors found on MDSCs and APCs. Toll-like receptor 9 is likely to prime various immune cells to promote antitumour T-cell function [80,81]. A Phase 1b trial (NCT05220722) is investigating the safety of SD-101 in combination with systemic single- (pembrolizumab) or dual-agent (nivolumab–ipilimumab) checkpoint blockades in patients with primary liver cancer.

The exhaustion of T cells can also prevent antagonizing the binding between galectin-9 (expressed in tumour-infiltrating macrophages) and immune checkpoint TIM3. The first trial of the anti-TIM3 antibody LY3321367 in combination with an anti-PD1 agent (NCT03099109) was stopped due to unspecified reasons [78], while a second trial investigating a different anti-PD1/anti-TIM3 combination (dostarlimab + cobolimab) in HCC patients is still ongoing (NCT03680508).

Lastly, the strategy of creating a less immune-suppressive TME by avoiding tryptophan depletion has not shown satisfactory results, as the IDO1 inhibitor epacadostat in combination with pembrolizumab did not improve the outcome of melanoma patients, leading to an early discontinuation of the trial dedicated to HCC patients [82].

## 7. Future Perspectives

Expanding knowledge about the mechanisms of resistance to immunotherapy will be the fundamental pre-requisite to improve the treatment of malignancies. In the case of HCC, however, specific additional complexities emerge. First, most HCCs arise in cirrhotic livers. Currently, no data support the hypothesis that the cirrhotic microenvironment might affect the efficacy of ICIs. When the first immunotherapy trials for HCC were designed, this hypothesis (deriving from preclinical experiences) was considered but it was never demonstrated [83]. Second, the mechanisms of immune escape in HCC are peculiar. In contrast to the current dogma that immune suppression/evasion occurs either early, during carcinogenesis, or late, prior to metastasis, Nguyen et al. [84]. recently reported that in HCC it is a progressive and continual process peaking at the TNM (tumour size, lymph node involvement and metastasis) stage II. The authors also provided potential evidence of partial immune recovery in TNM stage III tumours, corresponding with an increase in neoantigens [84]. Therefore, exploiting neoantigen exposure and immune resurgence in stage III tumours might theoretically maximise the efficacy of ICIs in advanced-stage HCC. Whether this peculiar trajectory of immune evasion and cancer progression is related to liver cirrhosis or other factors remains to be elucidated.

Third, the role of aetiology of liver cirrhosis is another debated topic. It has been recently reported that HCCs developing in the background of non-alcoholic steatohepatitis (NASH) are less responsive to immunotherapy due to alterations in the immune components of the liver (in particular, an expanded proportion of exhausted CD8 + PD1+ T cells [85]. More recently, immunosuppressive CXCR2+ neutrophils have been associated with resistance to anti-PD1 immunotherapy in HCC patients with NASH, and CXCR2 antagonism resulted in a resensitisation to immunotherapy [86]. Until now, clinical outcomes from HCC aetiology have varied across immunotherapy studies, suggesting that effects of underlying liver disease and tumour characteristics are only partially understood [4,5,60,69]. A potential relationship between hystologic variants of HCC and response or resistance to immunotherapy has been suggested [87]. In particular, the 2019 World Health Organization classification mentions lymphocyte-rich HCC [88]. This rare variant (<1%) is characterised by an abundant lymphocyte infiltration and a high PD-L1 expression rate on surrounding inflammatory cells [87], suggesting a possible good response to ICIs [89]. Surely, more data about histologic variants might help to achieve a better understanding of response and resistance to ICIs.

Lastly, the most appropriate endpoint for trials exploring first-line therapy for HCC has yet to be defined. PFS is known for being an unreliable measure of clinical benefit in this cancer [90,91]. Only educated guesses can be made from this endpoint in Phase 2 trials [92], whereas its use as the sole primary endpoint in Phase 3 HCC trials is highly debated [92,93]. At the same time, OS is increasingly affected by the growing availability of sequential treatments based on ICIs and TKIs. This drawback has become particularly evident in several recent trials of immunotherapy in HCC (including CheckMate-459, COSMIC-312, HIMALAYA, and possibly LEAP-002), in which up to 40% of patients in the TKI comparator group received at least one post-trial treatment contributing to generating unprecedented OS in the control groups [5,60,69,70]. Therefore, these problems support the revaluation of response criteria for predicting overall survival in HCC.

Finally, the safety of novel drugs is another matter of concern in the fragile setting of cirrhotic patients, as combining multiple agents could result in reduced eligibility [94].

## 8. Conclusions

The mechanisms of immune resistance are manifold and strictly intertwined. Based on this fact, clinical trials are testing combinations of drugs that might overcome this resistance and improve the results of immunotherapy in HCC, similarly to what has been reported in other oncological and haematological malignancies. For this reason, most of the novel therapeutic agents are being tested in combination with the standard-of-care ICIs, rather than as standalone agents. Among these molecules, drugs targeting the LAG3 and PTEN-STAT3 pathways are the most promising in the HCC setting, being supported by solid preclinical evidence. A further expansion of knowledge on carcinogenic mechanisms is crucial to identify novel investigational drugs.

## Figures and Tables

**Figure 1 cancers-14-04616-f001:**
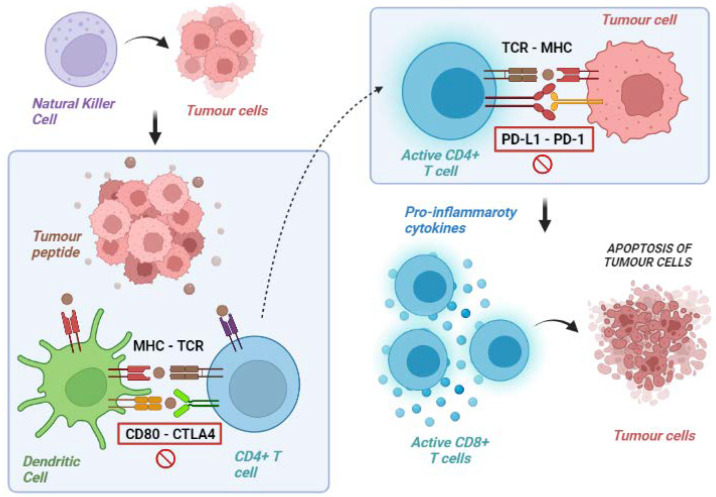
Exemplification of tumour immunity cycle. Natural killer and other cells of the immune system provoke lysis of tumour cells with liberation of tumour peptides. These peptides are exposed on surface molecules on dendritic and other antigen-presenting cells and are recognised by CD4 T lymphocytes (priming phase). These lymphocytes, therefore, acquire the ability to identify tumour cells and to promote a cytotoxic response via CD8 lymphocytes (effector phase). MHC: major histocompatibility complex; TCR: T-cell receptor; CD: cluster differentiation; CTLA-4: cytotoxic T lymphocyte-associated protein 4; PD-1: programmed death-1; PD-L1: programmed death 1-ligand.

**Figure 2 cancers-14-04616-f002:**
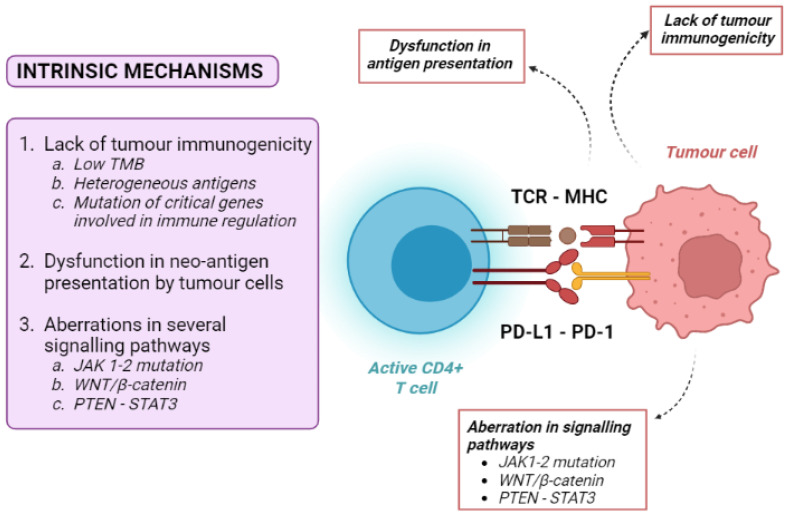
Overview of tumour-intrinsic mechanisms of resistance to immune checkpoint inhibitors. MHC: major histocompatibility complex; TCR: T-cell receptor; CD: cluster differentiation; PD-1: programmed death-1; PD-L1: programmed death 1-ligand.

**Figure 3 cancers-14-04616-f003:**
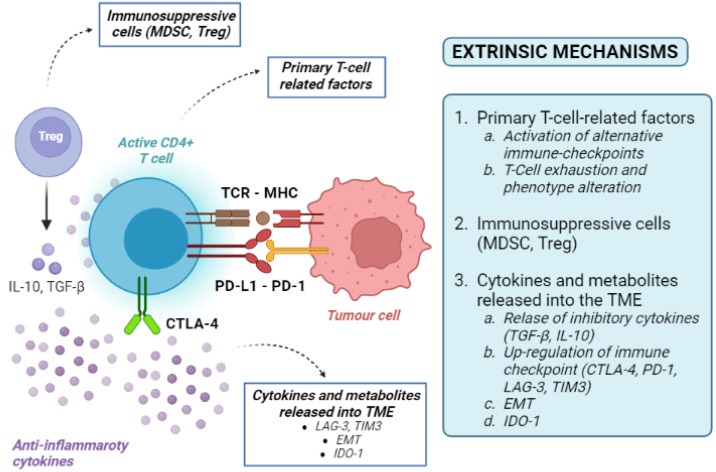
Overview of tumour-extrinsic mechanisms of resistance to immune checkpoint inhibitors. MHC: major histocompatibility complex; T-reg: regulatory T cells; TCR: T-cell receptor; CD: cluster differentiation; PD-1: programmed death-1; PD-L1: programmed death 1-ligand; MDSC myeloid-derived suppressor cells; IL-10: interleukin-10; CTLA-4: cytotoxic T lymphocyte-associated protein 4; TME: tumour microenvironment; EMT: epithelial-to-mesenchymal transition; TGF-β: transforming growth factor-beta; IDO-1: indoleamine-pyrrole 2,3-dioxygenase 1.

**Figure 4 cancers-14-04616-f004:**
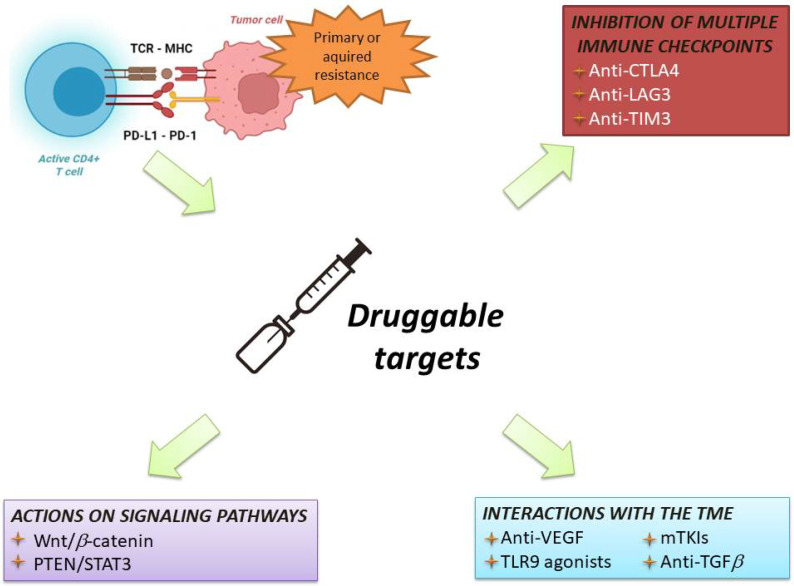
Overview of current therapeutic strategies to overcome resistance to immunotherapy. MHC: major histocompatibility complex; TCR: T-cell receptor; CD: cluster differentiation; PD-1: programmed death-1; PD-L1: programmed death 1-ligand. CTLA-4: cytotoxic T lymphocyte-associated protein 4; TME: tumour microenvironment; LAG3: lymphocyte-activation gene 3; TIM3: T-cell immunoglobulin and mucin domain 3; VEGF: vascular endothelial growth factor; TLR9: Toll-like receptor-9; mTKIs: multitarget tyrosin kinase inhibitors; TGFb: transforming growth factor-beta.

**Table 1 cancers-14-04616-t001:** Immune checkpoint inhibitors classified according to their main targets.

CTLA-4	PD-1	PD-L1	LAG-3
Ipilimumab	Nivolumab	Durvalumab	Relatlimab
Tremelimumab	Pembrolizumab	Avelumab	
	Camrelizumab	Atezolizumab	
	Dostarlimab		
	Toripalimab		
	Spartalizumab		
	Cempilimab		
	Sintilimab		
	Serpulimab		
	Nofazinlimab		
	Penpulimab		

CTLA-4: cytotoxic T lymphocyte-associated protein 4; PD-1: programmed death-1; PD-L1: programmed death 1-ligand; LAG3: lymphocyte activation gene-3.

## Data Availability

Not applicable.

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
