# Peer review of "Mechanisms of Primary and Acquired Resistance to Immune Checkpoint Inhibitors in Patients with Hepatocellular Carcinoma"

_cancers, 2022, doi:10.3390/cancers14194616_

Round 1

Reviewer 1 Report

The manuscript is well-organized and well-written.

There are some minor comments.

HCCs are classified into eight histologic variants in 2019 WHO classification. It would be better to add the differences in resistance mechanisms to immune checkpoint inhibitors between histologic variants of HCC (e.g., conventional HCC vs. lymphocytes-rich HCC) or between underlying diseases (e.g., viral hepatitis B, alcoholic).

It would be better to check English spelling.

 e.g.,  subsequenty activated antigen-specific response -> subsequently  

          activated antigen-specific response

         heterogeneus antigen -> heterogeneous antigen

Author Response

The attached file contains our point-to-point response

Author Response

The attached file contains our point-to-point response. Thanks
